# An Isolated Hydatid Cyst of the Spleen with High Serum Levels of CA 19-9—A Meaningful Association or Just a Challenge for Diagnosis? A Case Report

**DOI:** 10.3390/medicina61020182

**Published:** 2025-01-22

**Authors:** Traian Dumitrascu

**Affiliations:** Department of General Surgery, Division of Surgical Oncology, Fundeni Clinical Institute, Carol Davila University of Medicine and Pharmacy, Fundeni Street No 258, 022328 Bucharest, Romania; traian.dumitrascu@umfcd.ro

**Keywords:** hydatid cyst, spleen, high serum level of CA 19-9, splenectomy

## Abstract

An isolated hydatid cyst of the spleen represents an exceptional pathology, and its association with high CA 19-9 serum levels was not previously reported. This case presents a patient with an isolated hydatid cyst of the spleen, with preoperative high CA 19-9 serum levels in the absence of other pathologies and normalization of CA 19-9 serum levels after surgery (i.e., splenectomy). The source and clinical value of high serum levels of CA 19-9 in hydatid cysts of the spleen remains unclear. High serum levels of CA 19-9 in the context of a splenic cyst may complicate diagnosis and challenge the therapeutic strategy.

## 1. Introduction

Isolated hydatid cysts of the spleen are exceptional, and often reported as a case presentation or a small series of patients [1,2,3]. Surgery represents the main therapeutic option for these patients [1], with an emerging role of minimally invasive, parenchyma-sparing procedures [1,3,4,5]; however, total splenectomy is indicated for large cysts [2]. High serum levels of the CA 19-9 tumor marker are usually associated with malignancies, mainly pancreatic and biliary tract cancers [6,7]. Although high serum levels of CA 19-9 are also reported in benign pathologies [6,7], there are only a few reports on patients with benign non-parasitic splenic cysts with elevated CA 19-9 serum levels [8,9,10]. Furthermore, the clinical value of high serum levels of CA 19-9 in the context of benign spleen pathology remains unclear [8]. In light of this, the case of a patient with an isolated hydatid cyst of the spleen, with preoperative high serum levels of CA 19-9 in the absence of other pathologies is presented, with normalization of CA 19-9 serum levels after surgery.

## 2. Case Presentation

A 25-year-old female non-smoker, non-drinker patient was admitted to Fundeni Clinical Institute in Bucharest for left upper quadrant abdominal pain. The symptoms appeared a few days before the presentation and became more intense, making the patient present at the hospital in an emergency setting. The medical history of the patient was unremarkable. The clinical examination revealed no pathological changes, including a gynecological clinical examination that did not identify any significant genital pathology (corroborated by further imaging explorations). The laboratory findings were within normal limits (including serum levels of amylase and lipase), except for very high serum levels of CA 19-9—700 UI/mL (normal: <37 UI/mL). An initial abdominal ultrasound examination identified a large cystic lesion in the left upper quadrant of the abdomen. However, it did not identify a particular point of origin (suggested pancreas vs. spleen). Further contrast-enhanced computed tomography revealed a large cystic lesion of 9.7/13/10.6 cm occupying almost the entire spleen, with linear calcifications of the wall. The cystic lesion was well circumscribed, with iodophil thin septum inside and fluid content without any solid component inside. The abovementioned cystic lesion compressed without invading the surrounding organs such as the stomach, left hemi-liver, distal pancreas, left kidney, jejunal loops, and the celiac trunk (Figure 1), and no other abnormalities were observed, particularly at the level of the pancreas, biliary tract, or genital organs. After the administration of non-steroidal anti-inflammatory parenteral drugs, the abdominal pain was relieved, allowing for a planned surgery. With a preoperative diagnosis of a large symptomatic cyst of the spleen, the patient was submitted to surgery in November 2023. At laparotomy, an isolated hydatid cyst of the spleen was discovered, and a total splenectomy was performed (Figure 2), with an uneventful postoperative outcome; the patient was discharged on postoperative day 5. The pathology examination confirmed the diagnosis of a hydatid cyst (Figure 3). One month after surgery, the serum levels of CA 19-9 decreased to 41.8 UI/mL and normalized at two months (CA 19-9 serum level = 10.8 UI/mL). The patient underwent postoperative administration of oral anti-parasitic drugs (albendazole 400 mg per day, three weeks with one week pause, for three months). The patient’s clinical examination, laboratory tests (including serum level of CA 19-9), and ultrasound examination at 6 and 12 months after surgery did not reveal any pathological findings.

## 3. Discussion

Elevated serum levels of CA 19-9 in hydatic disease have been previously reported in a limited number of patients. Thus, two studies analyzing the data of 39 to 40 patients with liver hydatid disease identified an elevated serum level of CA 19-9 in up to 28.2% of the patients; however, only mildly elevated values were observed in these studies [11,12]. A few other case report papers showed patients with very high CA 19-9 serum levels associated with a hydatid cyst of the breast [13] or liver [14]. However, the reported cases of liver hydatid cysts and elevated serum levels of CA 19-9 have had associated jaundice with cholangitis for the most significant part of the patients [14]; high serum levels of CA 19-9 are not uncommon in patients with benign etiologies of jaundice, albeit the mechanism remains largely unclear. Remarkably, our patient presented very high levels of CA 19-9 in the absence of jaundice or other pathologies.

Pfister et al. propose, as an explanation for high serum levels of CA 19-9 in hydatid disease, the presence of a few substances that possibly bear the Lewis-a antigen, or with closely related structures that are recognized by the anti-CA 19-9 antibodies; these substances are presumed to originate either from the parasite, or as a response of the host to the infection (particularly from Echinococcus multilocularis) [11]. As was the case in our reported patient, resection of the hydatid cysts was followed by normalization of CA 19-9 serum levels [11,12], a situation that might sustain the relationship between the hydatid disease and high CA 19-9 serum levels, albeit there are no shreds of evidence to sustain the Pfister theory. It was suggested that the epithelial cells of the splenic cysts might produce tumor makers within the cysts that may further be secreted into the blood [8].

High serum levels of CA 19-9 were previously reported in patients with non-parasitic benign spleen cysts, particularly epidermoid cysts [8,9,10]. This association may increase the complexity of the diagnosis, complicating the distinction between benign and malignant spleen pathology or even malignant pancreatic pathology, which may impact the therapeutic strategy. In almost all cases reported in the literature, the normalization of serum CA 19-9 levels was obtained after surgery [8].

In the above-presented case, the clinical differential diagnosis included any other cause of intense abdominal pain in a young woman. Thus, a gynecological pathology was excluded by the gynecological clinical examination, corroborated the imaging explorations. This aspect is exciting, because a few frequent gynecological pathologies in young women, such as cystic teratomas, ovarian abscesses, or endometriosis, may be associated with increased CA 19-9 serum levels [7,15]. Furthermore, a biliary and pancreatic inflammatory pathology was excluded, given the absence of risk factors from the anamnesis, the normal serum levels of amylase and lipase, and the normal appearance of the pancreas and gallbladder at ultrasound examination and computed tomography.

The initial ultrasound examination identified a sizeable cystic lesion without a precise point of origin, suggesting it was in the pancreas or the spleen. Thus, the patient was considered to be tested for serum CA 19-9 levels before the computer tomography exploration, as a cystic pancreatic neoplasm could not be excluded at that time. Cystic pancreatic neoplasms, such as mucinous cysts, solid pseudopapillary tumors, or even neuroendocrine tumors with cystic transformation should be included in the imaging differential diagnosis of a pancreatic cystic lesion in a young woman. A meta-analysis published in 2016 has shown that serum levels of CA 19-9 are valuable in differentiating benign and malignant cystic neoplasms [16].

The imaging differential diagnosis of a pancreatic cyst with parietal calcifications may include the following benign and malignant pancreatic pathologies: mucinous pancreatic cystic neoplasms [17,18], solid pseudopapillary tumors [17,19,20], neuroendocrine pancreatic tumors with cystic transformation [17,19], calcified pseudocyst in the context of chronic pancreatitis [17,21], or even pancreatic adenocarcinoma with cystic changes [20].

The imaging differential diagnosis of a splenic cyst with parietal calcifications may include other benign pathologies besides hydatid cysts, such as epidermoid cysts, splenic pseudocyst, calcified granulomas of infectious etiologies, or even malignant splenic pathologies: calcified splenic metastasis, lymphoma, epithelioid hemangioendothelioma, and splenic sarcoma [22,23]. However, a recent meta-analysis exploring imaging predictors of malignancies in splenic lesions associated calcifications with higher chances of benign pathology [24].

Other benign causes of elevated serum levels of CA 19-9 may include hepatic diseases (alcoholic liver cirrhosis or hepatitis, drug-induced hepatitis, or chronic hepatitis B virus, etc.), pulmonary diseases, rheumatoid arthritis, polycystic renal disease, or endocrine diseases, including diabetes mellitus [7,25,26]. A recent large populational study of young adults has associated elevated serum levels of CA 19-9 with an increased risk of developing type 2 diabetes in men, but not in women [27].

Interestingly, patients with an epidermoid cyst of the left diaphragm or primary retroperitoneal mucinous cystadenoma associated with high serum levels of CA 19-9 were previously described [28,29] and could be included in the imaging differential diagnosis of the above-presented case.

The normalization of the CA 19-9 serum levels after splenectomy in our patient, without any other further pathologies identified in the postoperative surveillance, makes the hydatid cyst of the spleen the most likely cause of its high preoperative CA 19-9 serum levels. However, the patient should be closely clinically monitored, including using bioumoral markers and imaging.

## 4. Conclusions

High serum levels of CA 19-9 in the context of a splenic cyst, including a hydatid one, may complicate diagnosis and challenge the therapeutic strategy. It may be difficult to distinguish between a benign and malignant spleen pathology, or even a malignant pancreatic pathology. The source and clinical value of high serum levels of CA 19-9 in hydatid cysts of the spleen remain unclear.

## Figures and Tables

**Figure 1 medicina-61-00182-f001:**
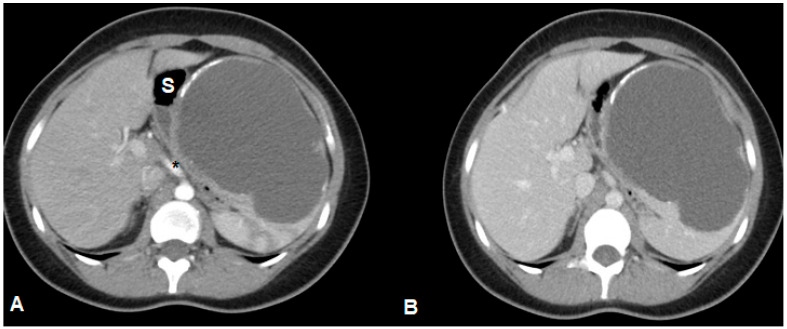
Contrast-enhanced computed tomography (axial plane, (**A**)—arterial phase; (**B**)—venous phase) shows a large cystic lesion of the spleen with wall calcifications, suggesting a spleen hydatid cyst, compressing the stomach (S) and the celiac trunk (*).

**Figure 2 medicina-61-00182-f002:**
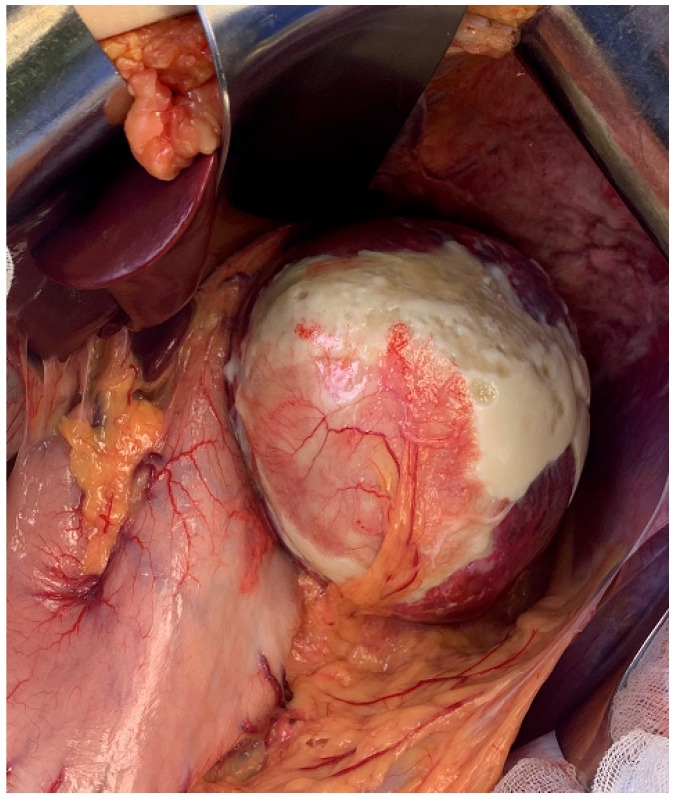
Intraoperative aspects show a large cystic lesion occupying the entire spleen with wall calcifications, suggesting a spleen hydatid cyst.

**Figure 3 medicina-61-00182-f003:**
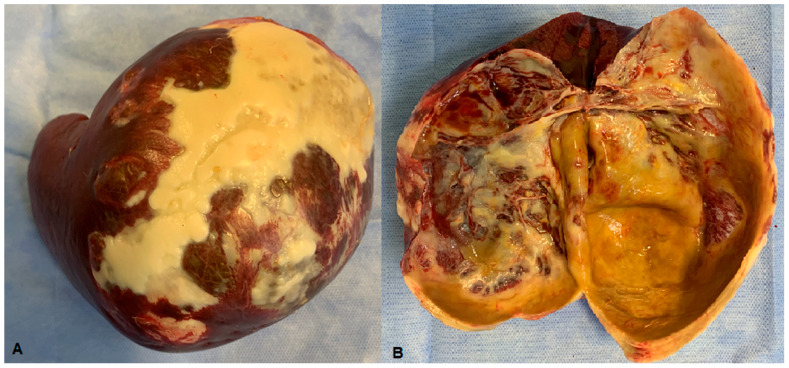
Operative specimen of splenectomy for hydatid cyst ((**A**)—entire operative specimen; (**B**)—transected operative specimen).

## Data Availability

All the data are presented in the manuscript.

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
