# Peer review of "An Isolated Hydatid Cyst of the Spleen with High Serum Levels of CA 19-9—A Meaningful Association or Just a Challenge for Diagnosis? A Case Report"

_medicina, 2025, doi:10.3390/medicina61020182_

Round 1
Reviewer 1 Report
Comments and Suggestions for Authors
Case Presentation:
I am not sure why you test for CA19-9?!! The image did not show any changes in either the biliary tract or the pancreas. In addition, the presence of calcification refers to hydatid cyst diagnosis.
Discussion:
-complicating the distinction between benign and malignant spleen pathology------->I disagree with you. The presence of a circumscribed cyst with calcified wall denotes that the lesion is benign. I suggest to change the ward spleen pathology to pancreatic pathology and remove the next sentence "or even malignant pancreatic pathology".
- It was suggested that the epithelial cells of the splenic cysts might produce tumor makers within the cysts that may further be secreted into the blood (8)-----------> This paragraph should be transferred to the previous paragraph as a second possibility to explain the huge elevation of CA19-9.
Conclusions:
- It may be difficult to distinguish between a benign and malignant spleen pathology or even a malignant pancreatic pathology-----> The same changes as in the discussion.
Author Response
Editorial comments: "We have just sent you the review reports. In addition to revising the paper based on the reviewers' comments, according to the requirements of the journal, the following point also needs to be revised: Enrich the contents: The length of the present version is a little shorter than what we expected for case report. In order to increase the readability of the article and to have a deeper understanding of the research content for readers, we are kindly suggesting you to add more details to support your research results when you revise your manuscript."
Response to Editorial comments: I have considered the Editorial suggestions and expanded the manuscript to enrich the contents, better understand the issues raised by the presented case report, and increase the manuscript's readability. The manuscript was modified accordingly.
Reviewer 1
Comment 1. "Case Presentation: I am not sure why you test for CA19-9?!! The image did not show any changes in either the biliary tract or the pancreas. In addition, the presence of calcification refers to hydatid cyst diagnosis."
Response to reviewer comment 1: The patient was tested for CA 19-9 serum level because the initial ultrasound examination could not exclude a pancreatic cystic neoplasm. The presence of calcifications is not exclusive to a hydatid cyst diagnosis. Details are provided in the revised version of the manuscript.
Comment 2. "Discussion: complicating the distinction between benign and malignant spleen pathology------->I disagree with you. The presence of a circumscribed cyst with calcified wall denotes that the lesion is benign. I suggest to change the ward spleen pathology to pancreatic pathology and remove the next sentence "or even malignant pancreatic pathology".
Response to reviewer comment 2: I'm afraid I disagree with the reviewer's opinion. Indeed, calcifications of a splenic cyst denote most likely a benign pathology. However, in rare cases, malignant spleen pathologies can exebit calcifications. Details are provided in the revised version of the manuscript.
Comment 3. "It was suggested that the epithelial cells of the splenic cysts might produce tumor makers within the cysts that may further be secreted into the blood (8)-----------> This paragraph should be transferred to the previous paragraph as a second possibility to explain the huge elevation of CA19-9."
Response to reviewer comment 3: I agree with the reviewer's suggestion, and the suggested change was made in the revised version of the manuscript.
Comment 4. “Conclusions: It may be difficult to distinguish between a benign and malignant spleen pathology or even a malignant pancreatic pathology-----> The same changes as in the discussion.”
Response to reviewer comment 4: Please consider the arguments provided to comment 2.
Reviewer 2 Report
Comments and Suggestions for Authors
Dear author,
congratulations for your work, is a quite interesting case but I think should be improved. Thereby I will adress here my suggestions:
- differential diagnosis: discuss the potential diagnoses considered and how the final diagnosis was determined
- investigations: detail any imaging, laboratory tests or other diagnostic procedures performed (was the CT scan the only diagnostic performed?)
- treatment: share more informations about your surgical steps and the medical treatment (which antiparasitic drugs were used)
Moreover I have some specific questions related to the clinical aspects:
- have you excluded any other CA 19-9 related condition? And how?
- Why was not considered to assess also CEA, CA 125, CA 15-3, TPA?
- what about the pancreatic status of the patient? Could an underlying acute pancreatitis be the cause of your CA 19-9 high levels? (Then labs related could be helpful)
- for such a young woman, was a gynaecological consult after 1 month considered or performed? In order to exlcude from endometriosis to dozens of ovary conditions
If your aim is to stress the possibile association between this cyst and those high CA 19-9 levels, it should be necessary to exclude at first other causes and even more try to picture that those high levels where isolated.
In conclusion, thank for your contribution and it would be nice if you can improve your paper following my suggestions.
Author Response
Editorial comments: "We have just sent you the review reports. In addition to revising the paper based on the reviewers' comments, according to the requirements of the journal, the following point also needs to be revised: Enrich the contents: The length of the present version is a little shorter than what we expected for case report. In order to increase the readability of the article and to have a deeper understanding of the research content for readers, we are kindly suggesting you to add more details to support your research results when you revise your manuscript."
Response to Editorial comments: I have considered the Editorial suggestions and expanded the manuscript to enrich the contents, better understand the issues raised by the presented case report, and increase the manuscript's readability. The manuscript was modified accordingly.
Reviewer 2
Comment 1. “Dear author, congratulations for your work, is a quite interesting case but I think should be improved. Thereby I will adress here my suggestions. In conclusion, thank for your contribution and it would be nice if you can improve your paper following my suggestions.”
Response to reviewer comment 1: Thank you for your comments, suggestions, and kind appreciation of the present manuscript. Your suggestions added value to the current manuscript.
Comment 2. “differential diagnosis: discuss the potential diagnoses considered and how the final diagnosis was determined”.
Response to reviewer comment 2: A paragraph addressing the differential diagnosis was added in the revised version of the manuscript.
Comment 3. “investigations: detail any imaging, laboratory tests or other diagnostic procedures performed (was the CT scan the only diagnostic performed?)”.
Response to reviewer comment 3: The requested information was added to the revised version of the manuscript. No other imaging investigation beyond the CT scan was considered necessary for management strategy.
Comment 4. “treatment: share more informations about your surgical steps and the medical treatment (which antiparasitic drugs were used)”.
Response to reviewer comment 4: The requested information was added to the revised version of the manuscript. There were no particular technical issues with the technique of open splenectomy to warrant further details.
Comment 5. “ have you excluded any other CA 19-9 related condition? And how?”.
Response to reviewer comment 5: We have excluded as much as we could other causes of elevated CA 19-9. Details are provided in the revised version of the manuscript.
Comment 6. “Why was not considered to assess also CEA, CA 125, CA 15-3, TPA?”
Response to reviewer comment 6: CEA, CA 125, and CA 15-3 were tested and they were within normal limits.
Comment 7. “what about the pancreatic status of the patient? Could an underlying acute pancreatitis be the cause of your CA 19-9 high levels? (Then labs related could be helpful)”.
Response to reviewer comment 7: An underlying acute pancreatitis was excluded based on the clinical, bioumoral, and imaging data. Details are provided in the revised version of the manuscript.
Comment 8. “for such a young woman, was a gynaecological consult after 1 month considered or performed? In order to exlcude from endometriosis to dozens of ovary conditions”.
Response to reviewer comment 8: A clinical gynecological examination corroborated with the imaging features excluded a significant genital pathology. Details are provided in the revised version of the manuscript.
Comment 9. “If your aim is to stress the possibile association between this cyst and those high CA 19-9 levels, it should be necessary to exclude at first other causes and even more try to picture that those high levels where isolated.”
Response to reviewer comment 9: The discussion was expanded to address this issue. Details are provided in the revised version of the manuscript.